# Cell Suspension Culture and In Vitro Screening for Drought Tolerance in Soybean Using Poly-Ethylene Glycol

**DOI:** 10.3390/plants10030517

**Published:** 2021-03-10

**Authors:** Nishi Mishra, Manoj Kumar Tripathi, Sushma Tiwari, Niraj Tripathi, Swapnil Sapre, Ashok Ahuja, Sharad Tiwari

**Affiliations:** 1Department of Plant Molecular Biology & Biotechnology, College of Agriculture, Rajmata Vijayaraje Scindia Agricultural University, Gwalior 474002, India; nishimishra554@gmail.com (N.M.); sushma2540@gmail.com (S.T.); ahujarrljm@gmail.com (A.A.); 2Directorate of Research Services, Jawaharlal Nehru Agricultural University, Jabalpur 482004, India; nirajtripathi@jnkvv.org; 3Biotechnology Centre, Jawaharlal Nehru Agricultural University, Jabalpur 482004, India; swapnil.spr@gmail.com (S.S.); stiwari@jnkvv.org (S.T.)

**Keywords:** soybean, drought, in vitro selection, callus culture, cell suspension culture, somaclone

## Abstract

Soybean (*Glycine max* (L) Merrill) is used in India mostly as a substantial fund of protein and oil, which makes the crop significantly important. Somaclonal variation has been researched as a base of additional variability for drought in soybean. In the present experiment calli/cell clumps/embryoids rose from immature and mature embryonic axis and cotyledons explants were exposed to different concentrations of polyethylene glycol (PEG6000). A discontinuous method proved to be superior as it permitted the calli/embryoids/cell clumps to regain their regeneration competence. A total of 64 (12.21%) plantlets of genotype JS335 and 78 (13.13%) of genotype JS93-05 were regenerated after four consequent subcultures on the selection medium with an effective lethal concentration of 20% PEG6000, and proliferated calli/embryoids/cell clumps were further subcultured on Murashige and Skoog regeneration medium supplemented with 0.5 mgL^−1^ each of α-napthalene acetic acid (NAA), 6-benzyladenine (BA) and Kinetin (Kn), 20.0 gL^−1^ sucrose and 7.5 gL^−1^ agar. Putative drought-tolerant plantlets were acquired from genotype JS93-05 (38) in more numbers compared to genotype JS335 (26). Random decamer primers confirmed the presence of variability between mother plants and regenerated plants from both the genotypes. Since these plantlets recovered from tolerant calli/embryoids/cell clumps selected from the medium supplemented with PEG6000, the possibility exists that these plants may prove to be tolerant against drought stress.

## 1. Introduction

Soybean (*Glycine max* (L) Merrill) in India is used as a significant source of protein and oil. It is in great demand, the ever-increasing demand is unsatiated and the national production is quite insufficient, mainly due to ever-decreasing planting areas. The enlargement of the production area of soybean will add to the production of the crop and is the only way to meet the demand for it. The enlargement of the production area of soybean may necessitate the inclusion of less favorable cropping areas i.e., drought-prone areas, and a targeted programme to reduce competition with other more profitable crops such as rice, corn and millet. The development of high-yielding soybean varieties may necessitate crop improvement programmes even in abiotically stressed circumstances, drought being one of them. Drought has been found to be responsible for low yields in soybean as well as for other crop production in dry areas in India. For the development of crop cultivars with increased tolerance to drought, breeding and biotechnological methods may play vital roles, and selected cultivars may be used to increase agricultural production in dry land areas [1].

Plant cell, tissue and organ culture approaches have been found suitable to understand the mechanism involved in the development of stress tolerance in plants. In vitro culture curtails environmental discrepancies due to defined nutrient media, controlled conditions and homogeneity of stress application. In addition, the straight forwardness of such manipulations enables studying large plant populations and stress treatments in a limited space and short span of time [2]. Reports are available on the in vitro selection of tolerant or resistant somaclones or mutants for biotic and abiotic factors, and selected plants have been used as donors of the targeted gene(s) in crop improvement [3]. It is now, a well-established fact that variations occur during cell division and differentiation in vivo [4]. Meristem cells, which serve as a ‘germ line’, are, by and large, immune to such genetic changes. In the normal life cycle of a plant, the mutant somatic cells are wiped out during sexual reproduction and are not passed on to the next generation. Such mutant cells, however, have the ability to divide and multiply (as do non mutant cells) when plant tissues are placed in a culture. Imposing selection pressures on cultured cells can result in favored growth of mutant cells, setting up mutant cell lines from which whole plants are recovered. The procedure is useful for the development of resistance/tolerance to certain pathogenic toxins, salt, drought, herbicides, anti-metabolites and heavy metals, and for screening mutants that produce useful amino acids in excess.

In soybean, reports are available on the application of in vitro selection techniques for different purposes such as sodium chloride-tolerant variants [5], herbicide resistance, increased methionine and proline content and drought tolerant cell lines [2]. Furthermore, different responses for drought tolerance were identified from somaclonal variants by Matheka et al. [6] and Mahmood et al. [7]. Somaclonal variation may be the result of a number of DNA changes and, in such cases; these changes are transmissible to the progeny generation. Phenotypic changes have also been observed in in vitro regenerated plants, and some of them may be utilized for the selection of superior plants against biotic and abiotic stresses [8].

Polyethylene glycol (PEG) of higher molecular weight has proved its role as an osmotic agent [9], and its application has been found to lower water potential [10,11]. Use of PEG or mannitol during in vitro selection for identification of abiotic stress-tolerant tissues has been reported earlier in potato and rice [8,12,13]. Various reports are available on use of PEG for successful in vitro selection for drought tolerance in maize [6], pelargonium [14], banana [15], sorghum [16], peanut [17] and soybean [2,18]. However, to achieve such a goal, a competent and reproducible plant regeneration method must be accessible from callus and cell suspension cultures. In our previous experiments, efficient and reproducible regeneration systems with similar or comparable genotypes have been established [19,20]. For this purpose, different explants such as mature embryonic axis [20,21], immature cotyledons [22], immature embryonic axis [23], leave discs [24] and hypocotyls [19,25] have been used previously with varying degree of success.

There has been very limited work done on germplasm line(s)/varieties of soybean cultivated in Madhya Pradesh and adjoining regions, which are well known for soybean production in the country (India) for drought-tolerance aspects. Therefore, there is great scope in the present research for discovering new genotypes that would enhance yield-contributing characters, as well as tolerance to drought by applying an in vitro selection approach.

## 2. Results and Discussion

In vitro selection proffers a vast prospective for the rapid and comprehensive production of useful somaclones or mutants for resistance/tolerance against various biotic and abiotic stresses including drought. These plants may serve as outstanding contributors of resistance genes in breeding programmes. During the present investigation, cell clumps/embryoids were exposed to PEG6000 to induce mutants to develop putative drought-tolerant plantlets. 

### 2.1. Raising of Callus Cultures

In the present investigation, for immature and mature embryonic axis and cotyledons cultures, considerable discrepancy among applied plant growth regulator type concentrations and their interactions with each genotype for each phase of cultures were evidenced (data not presented). Similar findings have been reported earlier for mature [20,26] and immature cotyledons [21] and for mature [20] and immature embryonic axis [23]. In the present study, for raising callus cultures, MS medium fortified with each of 5.0 mgL^−1^ 2,4-D and NAA in combination with 0.5 mgL^−1^ BA proved to be superior among all combinations tested. Similar findings were shown by Saepudin et al. [11] in soybean. Therefore, the above-mentioned combination was further considered for in vitro selection including different concentrations of PEG6000. However, for subsequent subculturing, a reduced level of 2,4-D (2.0–3.0 mgL^−1^ 2,4-D) in combination with 5.0 mgL^−1^ NAA and 0.5 mgL^−1^ BA was supported for faster development of somatic embryos. The results of preliminary experiments (auxins and cytokinins as sole) exhibited (Table 1) tremendous disparities regarding in vitro responses. A few combinations led to morphogenesis from cultured explants, while others produced only callus with higher or lower growth rates.

### 2.2. Establishment of Embryogenic Cell Suspension Cultures

Yellow to white-coloured embryogenic calluses, ~1.0 g in mass, friable in texture were transferred to liquid media (50 mL). Initially, it was observed that in liquid culture, calli could not be fragmented to form suspensions of cells or small cell clumps, which could possibly be due to higher lignification. Tiwari et al. [27] had similar observation for onion, Jhankare et al. [28] for withania, Bariwa et al. [29] for muskmelon, Uikey et al. [30] for *Rauvolfia serpentina*, Patidar et al. [31] for chitrak and Sharma et al. [32] for grape cell suspension cultures in liquid medium. To obtain small cell clumps, calluses were agitated instinctively. Friable calli when agitated, were smoothly broken and dispersed into clumps of ~2.0–5.0 mm. Further agitation splintered these clumps into small cell aggregates. Cultivar JS93-05 (Figure 1D) responded better compared to JS335 (Figure 1A) for higher cell clump formation leading to embryogenesis. To initiate a cell suspension culture, callus lines were chosen. However, during observation under the stereomicroscope most of the cells were found to be undifferentiated, with dense cytoplasm and prominent nuclei. Suspension cultures raised from embryogenic callus produced embryoids of different shapes and sizes and pursued diverse developmental pathways, the first pathway starting from the globular stage (Figure 1B,C,E,F), heart stage (Figure 1G) torpedo stage (Figure 1H) leading to the cotyledonary stage (Figure 1I,J). Another developmental pathway, followed by the cultured embryogenic cultures, was the proliferation of bipolar embryoids (Figure 1K) that enlarged and germinated after maturation (Figure 1L). In some instances, formation of multiple embryoids on primary embryoids (Figure 1M) was also evidenced. The stereomicroscopic observations demonstrated that the developing embryos were globular shaped 0.5–8 mm in size and yellow-green in colour. Individual clumps were observed to be multilobed, where each lobe represented an early-stage globular embryo (Figure 1M). Embryos were truly attached at their bases to the cultures, but phenotypic appearance indicated their origin from the surface of primary embryos. Similar observations have been reported during soybean embryogenic suspension culture by Finer [33]. Proliferation of primary embryos gave rise to concentric layering of the proliferating secondary embryos until the clumps enlarged, seceded into the liquid culture medium and progressively germinated and produced shootlets (Figure 1N). Some embryoids were found to be converted into multipolar structures (Figure 1O) i.e., exhibiting multiple gamogenesis from a single embryoid (Figure 1P).

The fortification of culture media with auxins and cytokines in isolation did not keep up the instigation of suspension cultures in soybean in early experiments (data not presented). However, various combination of 2,4-D and NAA with BA were tested subsequently (Table 2). As compared to the suspension cultures raised from embryogenic calli, the initiation and growth response was totally different with all culture media combinations. For instance, 2,4-D was found to be deleterious for the initiation of suspension cultures from immature embryos. Even after three to four subculturings, cell growth declined and the suspension became necrotic. Several subcultures with culture medium containing higher concentrations of 2,4-D resulted in declines of cell growth and plasmolysis of the cells, which was mainly due to the gradual increase in the absolute concentration of 2,4-D per cell [27].

Maximum relative growth rate was accomplished with culture media MS5D5N.5B followed by liquid medium MS5D10N.5B. For the establishment of embryogenic cell suspension cultures, the liquid media containing 5.0 mgL^−1^ each of 2,4-D and NAA in combination with a lower concentration of 0.5 mgL^−1^ BAP was found to be the most effective. Parallel results were addressed by Saepudin et al. [11] in soybean liquid culture. During the first two subculturings this media gave rise to loose, friable cell aggregates from the cultured embryogenic calli, which formed a fine and light cream and yellow cell suspension. For subsequent subculturing 2,4-D concentration was reduced to 2.0–3.0 mgL^−1^, which supported more rapid expansion of embryoids. 

### 2.3. Determination of the LD_50_ of PEG6000

In this experiment, PEG6000 was used as a selection agent. More than 90% mortality was observed at higher (30% *w/v*) PEG6000 concentration (Table 3; Figure 2G,N; Figure 3K,L). However, any significant negative effect on callus growth was not evidenced at lowest level of PEG6000 (5.0% *w/v*) concentration (Figure 2B,I and Figure 3B). At the level of 20% *w/v* PEG6000, about 50% callus growth rate was monitored (Figure 2E,L; Figure 3G,H). Thus, LD_50_ (20% *w/v*) PEG6000 concentration was used for selection criteria. The control callus (without addition of PEG 6000) as well as the callus grown up to the 15% (*w/v*) level of PEG6000 (Figure 2A–D,H–K; Figure 3A–F) looked healthy and survived considerably, whereas the calli treated with the highest PEG6000 level showed a brown-black colour and eventually died (25–30% *w/v* PEG6000; Figure 2F,G,M,N; Figure 3I–L). This may be due to the reduction of cytoplasmic and vacuolar volume because of the exclusion of water from the cytoplasm [34]. In such cases the cells may be unable to take up nutrients and water from their outer environments due to lowered external osmotic potential resulting in a decline in cell growth. Saepudin et al. [11] also reported that PEG6000 at the concentration of 15% *w/v* was effective for selecting tolerant calli in soybean. Related to that, the resulting freshly embryogenic callus of both the genotypes in our study might be somaclonal variants that have potential tolerance to drought.

Following completion of the selection cycle, numbers of tolerant calli/cell clumps/embryoids on 5% (*w/v*), 10% (*w/v*) and 15% (*w/v*) PEG6000 levels were not found to be significantly different. The calli and embryoids/cell clumps of both the genotypes survived up to a concentration of 20% PEG6000 and looked fresh but differed from 25% (*w/v*) and 30% (*w/v*) PEG6000 because of lower survival rate. The application of PEG stimulates drought because of its property to inhibit water in such a way that no water is provided for the somatic cell, except for the callus/somatic cell which has a particular mechanism for absorbing water. Addition of PEG in selective media causes dehydration stress. PEG stress tolerant embryogenic callus and embryoids could also be drought stress-tolerant somatic embryo variants [35]. Hence, soybean plants regenerated from PEG stress-tolerant embryogenic callus and cell clumps/embryoid should also be drought-stress tolerant.

### 2.4. In Vitro Selection

Two methods, continuous and discontinuous, were applied for performing selection against drought. Through the continuous process, 36.80% and 37.45% percentage survival rate of soybean genotypes JS335 and JS93-05 was found, respectively, after the first cycle of selection of callus cultures on medium containing 20% PEG6000 (Table 4). However, during the second cycle about 2–3% of the calli still died, but thereafter calli were subjected to further selections that exhibited insensitivity to medium with PEG6000, since a smaller number of calli died (1–2%). Selection by the continuous method resulted in callus survival rates 27.20% for JS335 and 29.09% JS93-05, respectively after fourcycles of selections. However, in the discontinuous method, in the first cycle of selection the rate of survival was 39.50% for JS335 and 41.07% for JS93-05, respectively. A break of two cycles was given by subculturing tolerant calli in maintenance medium. In conclusion, in the fourth cycle of selection 35.33% calli of genotype JS335 and 38% calli of genotype JS93-05 survived.

Application of the continuous method with cell suspension cultures resulted in 33.94% of surviving cell clumps of JS335 and 35.90% of JS93-05 after the first cycle of selection on the medium with supplementation of 20% (*w/v*) PEG6000 (Table 5). About, 2–3% of the calli died during the second cycle prior to acquiring insensitivity for the duration of the third cycle. Thereafter, only few embryoids/cell clumps (1–2%) died due to PEG6000 presence in the medium during the fourth selection. After four cycles of selection by the continuous method, the survival rate of embryoid/cell clumps for JS335 was 25.02% while for JS93-05 it was 28.09%, whereas in the discontinuous method the conclusive survival rates of embryoids/cell clumps were 32.40% and 34.47% for JS335 and JS93-05, respectively. Similar observations were recorded during in vitro selection experiments in wheat and barley [36], onion [37] and Withania [38].

The present investigation demonstrated callus mortality of 65–70% in both the genotypes during the first cycle of selection. Almost similar results, after completing four cycles of selection by continuous and discontinuous methods, were observed but the discontinuous method exhibited an apparent gain of high regeneration propensity over the continuous method. Similar findings have been reported during in vitro selection for wheat and barley by Chawla and Wenzel [36], for onion by Tripathi et al. [37] and for Withania by Jhankare et al. [38].

In this experiment, the callus and cell clumps/embryoids were monitored by browning; the calli turned brown then black and eventually died, predominantlyin genotype JS335, which had much greater callus transience (Figure 1A–H; Figure 2I–L). The results indicated the differences in survival to water stress could not be attributed to the genotypes. However, Lutts et al. [39], observed a genotype-dependent response to water stress induced by PEG or abscisic acid. In a study conducted by Mahmood et al. [40], PEG stress tolerant callus tended to prove less browning and had higher survival rates under high level of PEG6000. The results of the present investigation were in line with results of those studies on PEG stress tolerance in plants whichstated that by adding PEG6000 in the selective medium, callus proliferation and in vitro regeneration capacity were reduced in wheat and barley [36], potato [12], onion [37], wheat [41], tomato [42], Withania [38] and soybean [2,11,43,44].

Variations in PEG concentrations also affected the survival percentage of embryonic callus/cell clumps/embryoids. The results after three to four months in the selection medium showed that higher concentrations of PEG led to lower number of embryogenic callus in both genotypes. This finding was similar to the results of Widoretno et al. [45] and Saepudin et al. [11] where inhibition of growth and development of soybean explants with reduced number of somatic embryo/cell clumps were reported after application of PEG with selective media. The increased concentration of PEG in selective media deteriorated the somatic/callus embryos. Water stress by PEG in the media reduced the numbers of embryogenic callus of both the genotypes in this study. The decreased water potential by PEG not only exaggerated callus growth but also the capacity of cells in callus masses to form embryogenic cells with somatic embryos. The effects of PEG might be due to reduction in endogenous embryo differentiation [11,46]. The reason of this effect may be due to synthesis of inhibitory polyamine in explants under PEG stress [1,11]. Higher concentrations of PEG affected the development, proliferation and survival of embryoids. Perhaps PEG results in low osmotic pressure in the medium, and the ethylene oxide sub-units of PEG, retain water through the formation of hydrogen bonds as earlier suggested by Abdel-Raheem et al. [47], Sayar et al. [41] and Saepudin et al. [11]. This indicates less availability of water molecules in the selective medium supplemented with PEG6000. This lower osmotic pressure might be an inhibiting factor for proliferation of embryogenic callus and somatic embryos, resulting in either cytoplasmic or vacuolar volume reduction [48].

### 2.5. Regeneration of Plants

In totality 64 (12.21%) plantlets of genotype JS335 and 78 (13.13%) of genotype JS93-05 were regenerated from tolerant calli/embryoids/cell clumps acquired from selection medium after transferring them on MS regeneration medium amended with 0.5 mgL^−1^ each of NAA, BA and Kn, 20% PEG600, 20.0 gL^−1^ sucrose and 7.5 gL^−1^ Agar (Table 6; Figure 4A–G). Regenerants were subsequently transferred on rooting medium for in vitro root induction (Figure 4H,I). After transferring of putative drought-tolerant regenerants in the greenhouse (Figure 4J) followed by the polyhouse (Figure 4K,L) after hardening, twenty-six (40.65%) plants of genotype JS335 and thirty-eight (48.71%) plants of genotype JS335 survived (Table 6). The addition of PEG to the MS regeneration medium adversely affected the in vitro regeneration capability of the soybean cultivars. Comparable results were also documented by Sakthivelu et al. [2] and Saepudin et al. [11]. Matheka et al. [6] also reported delayed plantlet regeneration after PEG treatment that possibly may be attributed to the detrimental effects of selection on regeneration. The number of shoots per regenerating PEG-selected callus/cell clumps/embryoids was also reduced compared to the unselected one. Similar findings were reported by Lutts et al. [39], Matheka et al. [6] and Saepudin et al. [11]. A lower plantlet conversion rate also might be due to abnormalities of somatic embryos produced as suggested by Tereso et al. [46] and Saepudin et al. [11] who stated that PEG causes the reduction in yield of somatic embryos in addition to causing incomplete development and anatomical abnormalities of the somatic embryos. It appears that, in our results, PEG treatment, as well as genotypes, influenced regeneration capacity after selection (Figure 4F,G). Genotype JS93-05 (38) produced surviving plantlets in more numbers compared to genotype JS335 (26). Other reports also documented the strong influence of genotypes on regenerability under osmotic stress [11,39] and toxic culture filtrate/phytotoxin exposure [36,37,38,44] conditions.

Continuously proliferated and survived calli and cell clumps/embryoids for about three to four months yielded PEG6000 stress-tolerant calli and embryoids from PEG-sensitive standard soybean genotypes JS335 and JS93-05. Therefore, those calli and embryoids variants might gain PEG stress tolerant mechanisms during calli or embryoids/cell clump proliferation. Survival of calli and cell clumps under such selective medium might indicate that they were developed from mutant cells and tissues acquiring dehydration stress-tolerance mechanisms (Figure 4E–G). Regenerated plants from such tolerant calli might maintain the same mechanisms at the plant level and havethe same mechanisms causing drought tolerance in identified variant lines. Although PEG-induced dehydration stress may be different than drought stress in the field, the tolerance to both conditions might utilize comparable mechanisms such as maintaining high tissue water potential under stress. As a result, opting for calli/embryoids underneath PEG-induced stress may result in potential drought-tolerant variants.

### 2.6. Molecular Confirmation of Putative Drought Tolerant Plants

A total of ten Random Amplified Polymorphic DNA (RAPD) markers (Table 7) were used for the amplification of mother plants as well as selected putative drought-tolerant plants of soybean genotypes JS335 and JS93-05. Among all RAPD markers, OPN-13 was found to be able to produce polymorphic bands with template DNA of selected putative tolerant plants produced after considering JS335 as the mother genotype (Figure 5a). Primer OPN-13 amplified a polymorphic band sized 600 bp with all twenty-six putative tolerant plants selected from JS335. However, the absence of this polymorphic band in the donor genotype confirmed the presence of variability between the donor and the selected tolerant plants. Consequently, molecular variability between the mother genotype JS93-05 and selected plants was found to be putatively tolerant against drought after field evaluations were confirmed using two RAPD markers i.e., OPN-13 (Figure 5b) and OPM-12. Both of these markers allowed discrimination between the mother plant (JS93-05) and selected plants. Marker OPN-13 produced two polymorphic amplicons on 625 bp and 1000 bp which were amplified only with the template DNA of selected putative drought-tolerant plants, and both amplicons were absent in the mother plant (JS93-05). Consequently, the RAPD marker OPM-12 also produced polymorphisms between the template DNA of the mother plant (JS 93-05) and selected plants. An amplicon of 1000 bp was produced by marker OPM-12 in selected plants. However, this amplicon (1000 bp) was found to be absent in the mother plant (JS 93-05). Similarly, applications of RAPD markers for the same purposes have been reported by several authors [44,49,50] for the confirmation of the salt-tolerant somaclone of tomato plants.

## 3. Materials and Methods

### 3.1. Experimental Materials

Two drought-susceptible genotypes, JS335 and JS93-05, were used in the present investigation to develop tolerant soybean genotypes against drought. The seeds were acquired from All India Coordinated Research Project on Soybean, Rajmata Vijayaraje Scindia Krishi Vishwa Vidyalaya, Gwalior, Madhya Pradesh, India

Efficient and reproducible in vitro plant regeneration protocols from callus and cell suspension cultures were established before carrying out the in vitro cell line selection experiments employing PEG6000 as selection agent. Callus cultures were established by culturing mature and immature embryonic axis and cotyledons explants, whereas embryogenic cell suspension cultures were raised from embryogenic calli achieved from immature and mature embryonic axis and cotyledon explant cultures.

### 3.2. Culture Media

In an initial experiment, two different fortifications of basal media, MS [51] and B_5_ [52], were tested to optimize the in vitro response. During the initial stage of the investigation, MS basal medium was found to be more receptive than B_5_ medium (data not presented). Consequently, MS was used as basal medium for further work.

Apart from MS basal macro and micro salts, vitamins, and 7.5 gL^−1^ agar powder, two auxins, namely, 2,4-D and NAA, and a cytokinin BAP, in varying concentrations were added to fortify the MS media for culturing all the explants during initial experiments. Not all the cells within an explant, or the callus raised from them, went on to form organs or embryoids. Also, not all the explants reacted evenly to circumstances congenial for morphogenesis. It was seen that an auxin or a cytokinin alone, was not sufficient for inducing morphogenesis in higher frequencies. As a result, when deciding final experimental protocols, basal MS medium was fortified with both types of plant growth regulator (an auxin as well as a cytokinin) in varying concentrations and combinations.

For raising callus cultures, MS basal medium was fortified with 5.0 mgL^−1^ each of NAA and 2,4-D in combination with 0.5 mgL^−1^ BA, as this combination was found effectual in preceding studies conducted by various scientists, as well as the preliminary experiments of this laboratory. However, for establishment of cell suspension cultures, MS medium was supplemented with different concentrations and combinations of cytokinin BAP in combinations with two auxins, 2,4-D and NAA (Table 2). Readymade basal medium and all other ingredients were procured from Hi-media Laboratories, Mumbai, India. 

### 3.3. Establishment of Cell Suspension Cultures

For raising embryogenic cell suspension cultures, six weeks-old friable embryogenic calli acquired from immature and mature embryonic axis and cotyledon explant cultures were transferred into flasks containing 50 mL of liquid MS medium supplemented with different concentrations of 2,4-D and NAA in combinations with BA. A stainless-steel mesh (1 mm) was placed on the top of a funnel, the callus pieces were broken down by mashing with a spatula. 

### 3.4. Preparation of Culture Media

All initial culture media were made using readymade basal MS medium (HiMidia^TM^ Mumbai, India) and supplemented with different types of plant growth regulators in various concentrations and combinations with 30.0 gL^−1^ sucrose. The final volume was made to 1000 mL and pH was adjusted to 5.8 ± 0.1 with 1N KOH solution. After adjusting the pH, agar powder at 7.5 gL^−1^ or phytagel at 2.5 gL^−1^ (especially when PEG6000 was added in higher concentration in selection medium with callus cultures in semisolid medium) was added to the media as a solidifying agent. However, in the case of liquid media used for raising embryogenic cell suspension cultures, agar powder was not included. Warm culture media, still in a liquid state was poured into baby food bottles (50–60 mL/bottle) and culture tubes (15–20 mL/tube) followed by autoclaving at 121 °C under 15 psi pressure for 25 min. However, when pouring in petridishes, autoclaved warm culture media was poured into presterilized 100 × 17 mm glass petridishes (25–30 mL/dish) under aseptic conditions in a Laminar Flow Clean Air Cabinet.

### 3.5. Immature Embryonic Axis and Cotyledons Excision and Plating Technique

The explants, immature embryonic axes and cotyledons, were obtained from young field-grown plants. Immature embryonic axes were isolated from 14 to 21 days after anthesis, and seeds and immature cotyledons were obtained from 3 to 4 weeks-old post anthesis seeds. For this purpose, whole pods with immature seeds inside were surface sterilized for 2 min in 70% ethanol and then for 5 min in 1% mercuric chloride (HgCl_2_) solution followed by three subsequent rinsing with sterilized double distilled water. Immature embryonic axis and cotyledons were excised and cultured on 100 × 17 mm glass petridishes containing different inoculation media. In each petridish, 8–10 immature embryonic axes and 6–8 cotyledons were plated.

### 3.6. Mature Embryonic Axis and Cotyledons Excision and Plating Technique

The explants, mature embryonic axes and cotyledons, were taken from mature seeds. For this purpose, surface sterilization of seeds was performed by treating them with 70% ethyl alcohol for 1 min and then 5 min in 0.1% HgCl_2_ solution followed by three rinsings with sterilized double-distilled water. Surface sterilized seeds were imbibed in sterile double-distilled water for 24 h before utilizing them for isolation of embryonic axes. Mature embryonic axes were excised from the presoaked seeds and placed on different inoculation media. The mature cotyledons were obtained from germinating seedlings. For this purpose, surface sterilized seeds were inoculated in culture tubes containing agar gelled water (7.5 gL^−1^ agar) under diffused luminance of 16 μmol m^−2^s^−1^ provided with white, fluorescent lamps. Mature cotyledons were excised from four days-old germinating seedlings and were cultured on 100 × 17 mm glass petridishes containing different inoculation media.

### 3.7. Culture Conditions for Callus Cultures

Petridishes and baby food bottles containing cultures sealed with laboratory film (parafilm) were incubated under complete darkness at 25 ± 2 °C for one week. Later, in vitro cultured explants were subjected to a photoperiod regime each of 12 h light and dark at an intensity of 1200 lux luminance provided by white, fluorescent light.

### 3.8. Culture Conditions for Cell Suspension Cultures

The flask with cultures were agitated on a horizontal shaker at 120 rpm at 25 ± 2 °C, under a photoperiod regime each of 12 h light and dark at an intensity of 1200 lux luminance provided by white, fluorescent light. After 15 days the cultures were sieved aseptically to separate large clumps and 10.0 mL filtrate was added with 40.0 mL of fresh medium by replacing the old suspension for subculturing. Relative growth rate was calculated on the basis of change in fresh weight after culturing of embryogenic friable calli on different liquid media after 35 days of initial culture. Cell cultures were examined microscopically within 15 to 35 days for somatic embryoid/cell clump induction, and for deciding upon the developmental pathways. 

Relative growth rate was determined as follows:

Relative Growth Rate (%) = (Final weight − Initial weight) × 100

### 3.9. Determination of the LD_50_ of PEG6000

To determine the selection concentrations of PEG6000, calluses/embryoids/cell clumps were separated into small pieces and were placed on varying concentrations of PEG6000. LD_50_ was established with reference to approximately 50% retarded growth of callus embryoids/cell clumps. Fresh weight and relative growth rates of callus and cell suspension culture atseven different levels of PEG6000 i.e., 0% (control), 5, 10, 15, 20, 25 and 30% (*w/v*) were determined after 35 days.

### 3.10. In Vitro Selection and Regeneration Procedures

For initiation and selection of drought tolerance, small pieces of embryogenic calluses and cell suspension cultures were subjected to fortified MS culture medium supplemented with an LD_50_ dose of PEG6000 and 5.0 mgL^−1^ each of NAA, 2,4-D and 0.5 mgL^−1^ BA. Pursuing continuous and discontinuous selection methods, calli/cell clumps/embryoids were grown on MS medium with an effective concentration of PEG6000 (as determined by LD_50_) to select the putative drought tolerant cell lines.

### 3.11. Regeneration of Plants from Tolerant Calli/Embryoids/Cell Clumps

Tolerant calli/embryoids/cell clumps acquired from in vitro selection were transferred into regeneration medium which was basal MS medium fortified with 0.5 mgL^−1^ each of NAA, BA and Kn, with an effective concentration of PEG6000(as determined by LD_50_), 20.0 gL^−1^ sucrose and 7.5 gL^−1^ agar powder. The same was suggested earlier by Tripathi and Tiwari [26], Tripathi and Tiwari [21,22,23,24] and Tiwari and Tripathi [53]. Cultures were subjected to 25 ± 2 °C temperature and photoperiod regimes of 60 μmol m^−2^s^−1^ luminance provided by cool fluorescent tubes for 16 h after 35 days of culture. 

### 3.12. Rooting of Regenerants

Regenerants were further transferred to MS rooting medium supplemented with 1.0 mgL^−1^ IBA, 15.0 gL^−1^ sucrose and 7.5 gL^−1^ agar powder as suggested by Tripathi and Tiwari [21,22] and Tiwari and Tripathi [53].

### 3.13. Hardening of Regenerated Plants

The plantlets were uprooted from cultures and were systematically washed with running tap water to get rid of the unwanted agar and were planted in 2.5 cm root trainers filled with 1:1:1 sand, soil and farm yard manure (FYM) sterilized mixture. Root trainers with transplanted plants were placed under 30 ± 2 °C and 65 ± 5% RH for 15–30 days for adaptation. Acclimatized plants were then transferred to a net house for 30 days for hardening before shifting them to the field for transplantation.

### 3.14. Molecular Confirmation of Putative Drought Tolerant Plants(s) Using RAPD Markers:

DNA extraction of both of the mother genotypes, JS335 and JS93-05, as well as selected putative drought-tolerant plants was done using Qiagen DNA extraction kit following the manufacturer’s instructions. Extracted DNA samples were quantified by a nanodrop spectrophotometer and diluted up to 25 ng/µL. For PCR amplifications, a total of ten random decamer primers (Table 7) were used. The polymerase chain reaction (PCR) mixture consisted of 50 ng genomic DNA, 10 pmol primer, 200 μM of each dNTP and 1 unit of *Taq* DNA polymerase with PCR buffer supplied (TrisHCl, pH 9.0; 15 mM MgCl_2_). Cycling parameters were 45 cycles of 1 min at 94 °C, 1 min at 36 °C, 2 min at 72 ºC with a final extension time of 5 min at 72 °C. Amplicons were separated by electrophoresis on 1.5% agarose gel and visualized under a gel documentation system after staining with ethidium bromide.

## 4. Conclusions

In the present work, the methods of in vitro selection using PEG as a selection agent from two drought-susceptible soybean genotypes were developed for generating putative drought-tolerant somaclones. Plantlets were regenerated from putative drought-tolerant somaclones of both the genotypes JS335 and JS93-05. This system might also be applied to other soybean genotypes to obtain drought tolerant regenerants. In conclusion, the selection strategy could be utilized for exploitation of in vitro-induced variability for development of drought-tolerant varieties in soybean by employing conventional or molecular breeding approaches. Tolerant variants may be possible sources of drought tolerance.

## Figures and Tables

**Figure 1 plants-10-00517-f001:**
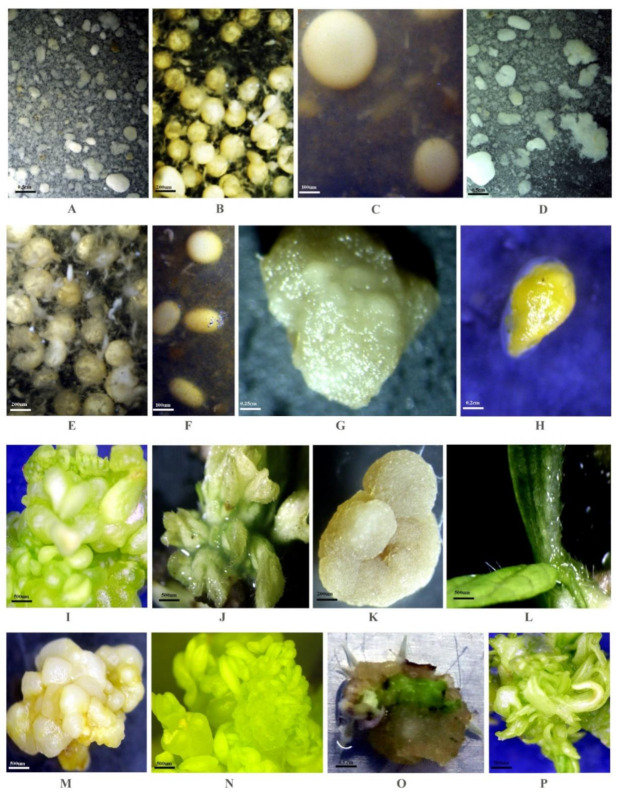
Plant regeneration from cell suspension culture in soybean: (**A**) Initiation of cell clumps and embryoid formation derived from embryogenic calli of genotype JS335. (**B**,**C**) Globular stage embryoid of genotype JS335. (**D**) Initiation of cell clumps and embryoid formation derived from embryogenic calli of genotype JS93-05. (**E**,**F**) Globular stage embryoid of genotype JS93-05. (**G**) Heart stage. (**H**) Torpedo stage. (**I**,**J**) Cotyledonary stage. (**K**) Typical bipolar embryoid. (**L**) Germination of bipolar embryoid. (**M**) Multiple embryoid. (**N**) Germination of multiple embryoid. (**O**) Typical multi-polar embryoid. (**P**) Germination of multipolar embryoid.

**Figure 2 plants-10-00517-f002:**
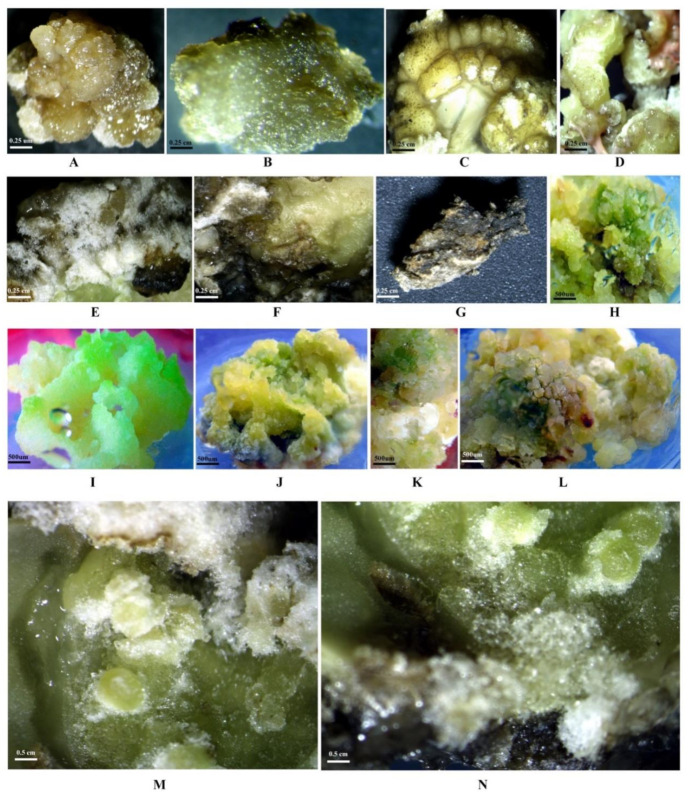
Effect of different concentrations of PEG6000 on survival of calli and formation of nonembryogenic/nonregenerable (**A**–**G**) and embryogenic/regenerable (**H**–**N**) callus in selection medium in soybean. (**A**) PEG6000 0% (*w/v*); (**B**) PEG6000 5% (*w/v*); (**C**) PEG6000 10% (*w/v*); (**D**) PEG6000 15% (*w/v*); (**E**) PEG6000 20% (*w/v*); (**F**) PEG6000 25% (*w/v*); (**G**) PEG6000 30% (*w/v*); (**H**) PEG6000 0% (*w/v*); (**I**) PEG6000 5% (*w/v*); (**J**) PEG6000 10% (*w/v*); (**K**) PEG6000 15% (*w/v*); (**L**) PEG6000 20% (*w/v*); M.PEG6000 25% (*w/v*); (**N**) PEG6000 30% (*w/v*).

**Figure 3 plants-10-00517-f003:**
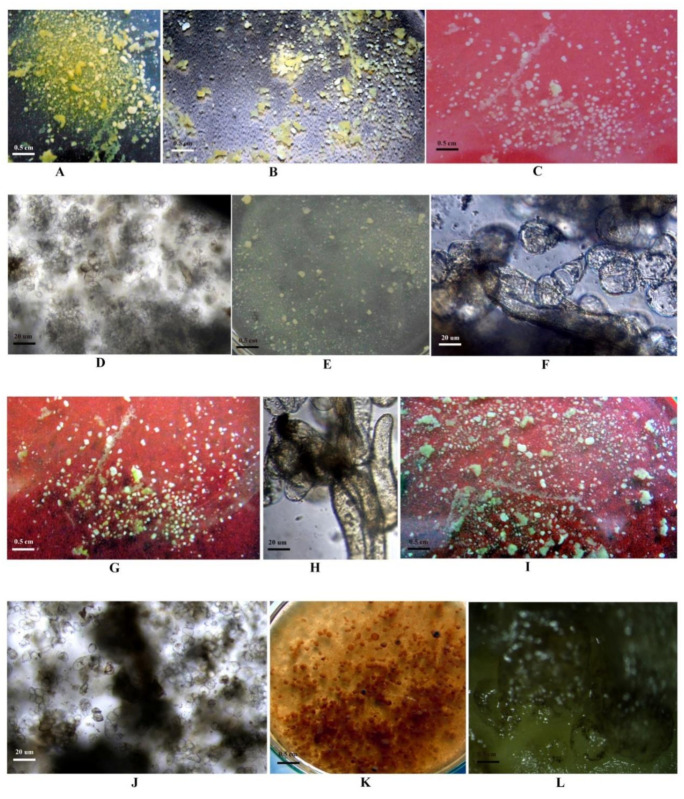
Effect of different concentrations of PEG6000 on survival of clumps/embryoids in liquid selection medium in soybean. (**A**) PEG6000 0% (*w/v*); (**B**) PEG6000 5% (*w/v*); (**C**,**D**) PEG6000 10% (*w/v*); (**E**,**F**) PEG6000 15% (*w/v*); (**G**,**H**) PEG6000 20% (*w/v*); (**I**,**J**) PEG6000.

**Figure 4 plants-10-00517-f004:**
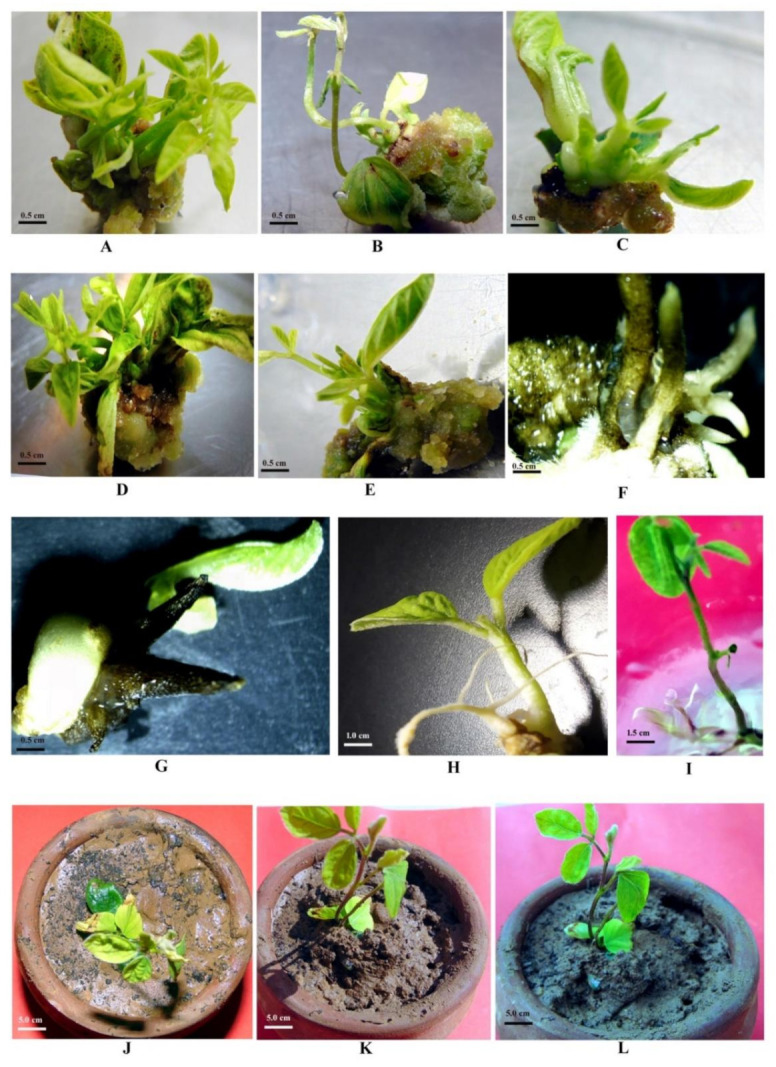
Effect of different concentrations of PEG6000 on regeneration of plantlets obtained from tolerant calli/cell clumps/embryoids in soybean. (**A**) PEG6000 0% (*w/v*); (**B**) PEG6000 5% (*w/v*); (**C**) PEG6000 10% (*w/v*); (**D**) PEG6000 15% (*w/v*); (**E**) PEG6000 20% (*w/v*); (**F**) PEG6000 25% (*w/v*); (**G**) PEG6000 30% (*w/v*); (**H**,**I**) Rooting of regenerants; (**J**) Regenrant transferred in greenhouse for hardening;(**K**,**L**) Regenerants transferred in nethouse for hardening.

**Figure 5 plants-10-00517-f005:**
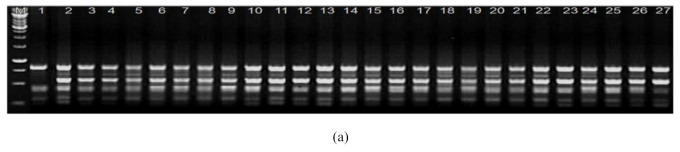
(**a**) Electrophoretic banding pattern of 1:JS335 and 2 to 27 selected putative drought tolerant plants amplified with Random Amplified Polymorphic DNA primer OPN-13. (**b**) Electrophoretic banding pattern of 1:JS93-05 and 2 t0 39 selected putative drought tolerant plants amplified with RAPD primer OPN-13.

**Table 1 plants-10-00517-t001:** Initial responses of immature and mature embryonic axis and cotyledon cultured on MS medium supplemented with different plant growth regulators in varying concentrations.

S. No	Culture Media	Plant Growth Regulator mgL^−1^			Direct Shoot Formation	Root Formation	Callus Formation	Non Morphogenic Friable Calli	Callus Colour	Callus Texture and Morphology
		NAA	2,4-D	BAP						
1.	MSN	1.0	-	-	-	+	+	-	Yellow greenish	Soft friable
2.	MS2N	2.0	-	-	-	++	++	-	Yellow greenish	Soft friable
3.	MS3N	3.0	-	-	-	++	++	-	Yellow greenish	Soft friable
4.	MS4N	4.0	-	-	-	+	+++	-	A mixture of yellow, green and whitish calli	Compact
5.	MS5N	5.0	-	-	-	+	+++	-	A mixture of yellow, green and whitish calli	Compact
6.	MS6N	6.0	-	-	-	+	++++	-	Yellow/green	Compact. Large in size
7.	MS8N	8.0	-	-	-	+	++++	+	Yellow/green	Compact. Larger in size
8.	MS10N	10.0	-	-	-	+	++++	++	Dark yellow	Compact. Largest in size
9.	MSD	-	1.0	-	-	+	++	+	Yellowish	Soft friable. Callus covered with white boundaries
10.	MS 2D	-	2.0	-	-	+	+++	+	Yellowish	Soft friable. Callus covered with white boundaries
11.	MS 5 D	-	5.0	-	-	+	++++	+	Yellowish	Soft friable. Callus covered with white boundaries
12.	MS10D	-	10.0	-	-	Rare	++++	++	Yellowish	Soft friable. Callus covered with white boundaries
13.	MS20D	-	20.0	-	-	Rare	++++	+++	Yellowish/white	Covered with dense layer of white loose calluses
14.	MS30D	-	30.0	--	-	Very rare	++++	+++	Yellowish/white	Covered with dense white loose calluses. Comparatively large in size
15.	MS40D	-	40.0	-	-	Very rare	++++	+++	Yellowish/dark brown	Covered with dense layer of brown loose calluses. Largest in size
16.	MSB	-	-	1.0	+	+	+	-	Green	Compact. Covered with white layers of loose calluses
17.	MS2B	-	-	2.0	+	Rare	++	-	Green	Compact. Covered with white layers of loose calluses
18.	MS3B	-	-	3.0	++	Rare	+++	-	Green	Compact. Covered with white layers of calluses
19.	MS4B	-	-	4.0	++	Very rare	+++	-	Green whitish	Compact. Covered with white layers of calli
20.	MS5B	-	-	5.0	-	Very rare	+++	-	Green. After 2–3 weeks of initial culture colour of calli changed in black	Compact. Covered with white layers of calluses

**Response:** Excellent: >75% (+ + + +); High: 50–75% (+ + +); moderate: 25–50% (+ +); low: <25% (+).

**Table 2 plants-10-00517-t002:** Combined effects of added cytokinins and auxins in varying concentrations and combinations on proliferation of embryogenic cell suspension cultures.

Genotype 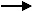 Medium 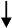	Plant Growth Regulators mgL^−1^	Cell Suspension Cultures
Embryonic Axis Derived Calli	Cotyledon Derived Calli
2,4-D	NAA	BA	JS335	JS93-05	JS335	JS93-05
Increment in FW (g) *	RG(%)	Increment in FW (g) *	RG(%)	Increment in FW (g) *	RG(%)	Increment in FW (g) *	RG(%)
**Control**	-	-	-	1.00 ± 0.08	100	1.08 ± 0.10	108	1.02 ± 0.08	102	1.10 ± 0.10	110
MSDN.5B	1.0	1.0	0.5	1.12 ± 0.14	112	1.18 ± 0.16	118	1.20 ± 0.20	120	1.28 ± 0.22	128
MS2DN.5B	2.0	1.0	0.5	1.24 ± 0.18	124	1.33 ± 0.20	133	1.30 ± 0.18	130	1.40 ± 0.22	140
MS3DN.5B	3.0	1.0	0.5	1.46 ± 0.24	146	1.52 ± 0.24	152	1.68 ± 0.28	168	1. 84 ± 0.28	184
MS4DN.5B	4.0	1.0	0.5	1.62 ± 0.26	162	1.74 ± 0.27	174	1.77 ± 0.29	177	1.92 ± 0.29	192
MS5DN.5B	5.0	1.0	0.5	1.94 ± 0.30	194	208 ± 0.32	208	2.12 ± 0.34	212	2.26 ± 0.26	226
MS10DN.5B	10.0	1.0	0.5	**CM**	-	**CM**	-	**CM**	-	**CM**	-
MS2D2N.5B	2.0	2.0	0.5	1.42 ± 0.20	142	1.53± 0.22	153	1.47 ± 0.22	147	1.58 ± 0.24	158
MS3D2N.5B	3.0	2.0	0.5	1.67 ± 0.26	167	1.74 ± 0.26	174	1.83 ± 0.30	183	1. 98 ± 0.31	198
MS4D2N.5B	4.0	2.0	0.5	1.85 ± 0.28	185	1.92 ± 0.29	192	1.96 ± 0.31	196	2.22 ± 0.32	222
MS5D2N.5B	5.0	2.0	0.5	2.07 ± 0.30	207	2.12 ± 0.31	212	2.16 ± 0.33	216	2.41 ± 0.34	241
MS10D2N.5B	10.0	2.0	0.5	**CM**	-	**CM**	-	**CM**	-	**CM**	
MS2D3N.5B	2.0	3.0	0.5	154 ± 0.24	154	1.72± 0.24	172	1.66 ± 0.24	166	1.79 ± 0.26	179
MS4D3N.5B	4.0	3.0	0.5	2.04 ± 0.30	204	2.15 ± 0.32	215	2.18± 0.33	218	2.42 ± 0.34	242
MS5D3N.5B	5.0	3.0	0.5	2.16 ± 0.34	216	2.30 ± 0.36	230	2.37 ± 0.35	237	2.52 ± 0.38	252
MS10D3N.5B	10.0	3.0	0.5	**CM**		**CM**		**CM**	-	**CM**	-
MS2D5N.5B	2.0	5.0	0.5	169 ± 0.26	169	1.83± 0.26	183	1.87 ± 0.26	187	2.01 ± 0.28	201
MS3D5N.5B	3.0	5.0	0.5	1.88 ± 0.28	188	1.96 ± 0.28	196	2.04 ± 0.32	204	2.21 ± 0.34	221
MS5D5N.5B	5.0	5.0	0.5	2.43 ± 0.36	243	2.54 ± 0.38	254	2.68 ± 0.39	268	2.81 ± 0.40	281
MS10D5N.5B	10.0	5.0	0.5	**CM**	-	**CM**	-	**CM**	-	**CM**	-
MS5D10N.5B	5.0	10.0	0.5	2.26 ± 0.32	226	2.41 ± 0.33	241	2.44 ± 0.37	244	2.62 ± 0.36	262

Evaluation was made after 35 days in culture. Initial fresh weight was taken 1.0 g per flask containing 50 mL liquid medium. **FW**: Fresh Weight; **RG**: Relative growth; **CM**: Cell Mortality. ***** Mean of five readings ± standard deviation.

**Table 3 plants-10-00517-t003:** Comparison of growth rates ^1^ of mature and immature embryonic axis and cotyledons derived calli and embryogenic cell suspension cultures on different levels of PEG6000.

Concentration of PEG6000 (% *w*/*v*)	Relative Growth Rate of Callus Cultures ^1^		Relative Growth Rate of Suspension Cultures ^2^	
	JS335	JS93-05	JS335	JS93-05
0.0	100.00 ± 0.38	100.00 ± 0.40	100.00 ± 0.36	100.00 ± 0.39
5.0	85.34 ± 0.32	87.91 ± 0.30	80.52 ± 0.34	88.78 ± 0.34
10.0	65.13 ± 0.26	68.48 ± 0.28	69.86 ± 0.30	75.86 ± 0.32
15.0	63.52 ± 0.24	65.68 ± 0.25	61.52 ± 0.26	66.34 ± 0.29
20.0	51.12 ± 0.22	52.19 ± 0.24	49.18 ± 0.24	52.40 ± 0.26
25.0	21.98 ± 0.20	23.78 ± 0.22	22.56 ± 0.22	25.84 ± 0.20
30.0	9.75 ± 0.16	11.54 ± 0.18	12.90 ± 0.16	14.62 ± 0.18

Mean was obtained from weights of five inoculums/treatments after 35 days of initial culture. ^1,2^ Callus and cell suspension cultures on MS medium fortified with each of 5.0 mgL^−1^ 2, 4-D, NAA, 0.5 mgL^−1^ BA and different levels of PEG6000.Callus and cell growth appeared average fresh weight.

**Table 4 plants-10-00517-t004:** Response of mature and immature embryonic axis and cotyledons derived calli of soybean to the lethal concentration of PEG6000.

Cultivar	Number of Calli Cultured	Number of Surviving Calli after 4 Selection Cycles			
		I	II	III	IV
***Continuous method***					
**JS335**	500	184 (36.80%)	163 (32.60%)	148 (29.60%)	136 (27.20%)
**JS93-05**	550	206 (37.45%)	187 (34.00%)	169 (30.72%)	160 (29.09%)
***Discontinuous method***					
**JS335**	600	237 (39.50%)	-	*-*	212 (35.33%)
**JS93-05**	650	267 (41.07%)	-	-	247 (38.00%)

Calli were cultured on selection medium (MS medium fortified with 5.0 mgL^−1^ 2, 4-D, 5.0 mgL^−1^ NAA, 0.5 mgL^−1^ BA and 20% (*w/v*) PEG6000).

**Table 5 plants-10-00517-t005:** Response of cell clumps/embryoids obtained from embryogenic cell suspension culture of soybean to the lethal concentration of PEG6000.

Cultivar	Number of Cell Clumps/Embryoids Cultured	Number of Surviving Cell Clumps/Embryoids after Selection Cycles			
		I	II	III	IV
***Continuous method***					
**JS335**	975	331 (33.94%)	297 (30.46%)	271 (27.79%)	244 (25.02%)
**JS93-05**	1025	368 (35.90%)	334 (32.58%)	306 (29.85%)	288 (28.09%)
***Discontinuous method***					
**JS335**	1000	345 (34.50%)	-	*-*	324 (32.40%)
**JS93-05**	1050	397 (37.80%)	-	-	362 (34.47%)

Calli/cell clumps/embryoids were cultured on **selection medium** (liquid MS medium fortified with 5.0 mgL^−1^ 2, 4-D, 5.0 mgL^−1^ NAA, 0.5 mgL^−1^ BA and 20% (*w/v*) PEG6000).

**Table 6 plants-10-00517-t006:** Regeneration frequency and survival percentage of soybean tolerant calli/cell clumps/embryoids against drought.

Genotypes	Calli/Cell Clumps/Embryoids	Plants Regenerated	Survival after Hardening in Poly House
**JS335**	524	64 (12.21%)	26 (40.65%)
**JS93-05**	594	78 (13.13%)	38 (48.71%)

Calli/cell clumps/embryoids were cultured on **regeneration medium** (MS medium fortified with each of 0.5 mgL^−1^ NAA, BAP and kinetin, 200 gL^−1^ sucrose and 7.5 gL^−1^ agar powder).

**Table 7 plants-10-00517-t007:** Details of RAPD primers used for confirmation of selected putative drought tolerant plants.

Primer	Sequence 5’-3’	JS335		JS93-05	
		Total Bands	Polymorphic Bands	Total Bands	Polymorphic Bands
OPA-5	AGGGGTCTTG	4	0	4	0
OPA-8	GTGACGTAGG	5	0	4	0
OPC-10	TGTCTGGGTG	4	0	5	0
OPC-15	GACGGATCAG	6	0	5	0
OPAP-07	ACCACCCGCT	6	0	5	0
OPAP-13	TGAAGCCCCT	3	0	3	0
OPR-15	GGACAACGAG	2	0	2	0
OPM 05	GGGAACGTGT	4	0	3	0
OPM-12	CTGGGCAACT	7	0	9	3
OPN 13	GGTGGTCAAG	6	3	7	3

## Data Availability

The data presented in this study is available within this article.

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
