# Peer review of "Cell Suspension Culture and In Vitro Screening for Drought Tolerance in Soybean Using Poly-Ethylene Glycol"

_plants, 2021, doi:10.3390/plants10030517_

Round 1

Reviewer 1 Report

The authors present a very interesting manuscript about soybean and the manuscript is clarly written.

The experiments are clearly explained.

Please add some more sentences in the conclusions summarizing your findings. 

Reviewer 2 Report

“Cell suspension cultures and in vitro selection for drought tolerance in soybean (Glycine max (L) Merr.) by applying Poly-Ethylene Glycol” by Mishra et al.

I agree that techniques of somaclonal variation are still crucial for breeding crops. However, I believe the authors still had many things to do carefully before the submission. There are too many such mistakes in format. Many typing errors. For instance, “mg.l-1” Why put “.” in between. In the pictures, there are no scale bars. There are too many unnecessary spaces, and a lacks of necessary space. Overall, this manuscript is too wordy and difficult to read. 

Data seem to be unnecessarily bulky. Contents seem to lack objectivity.

I cannot recommend this manuscript for publication.

Reviewer 3 Report

In general presented article is actual and informative, being devoted to the creation of drought tolerant soybean breeding initial material. Though the investigation was focused only on the utilization of in vitro culture techniques, thus selecting mutants, which drought tolerance is based only on the resistance to the dehydration stress in tissue culture, while intact plant uses many different mechanisms of resistance, the obtained results will contribute to the development of soybean breeding techniques and develop theoretical understanding of the possible ways of problem solution.

Article is logically structured and written in good English language. At the same time, there are some mistakes:

Line 46. ... mechanism INVOLVED in the development...

Line 78. For this purpose they were used different explants such as... Something is wrong in this sentence.

Line 169-170. Hence, soybean plants regenerated from PEG stress TOLERANT embryogenic callus and cell clumps/embryoid should also be drought stress TOLERANCT.

Line 236-237. ...thirty-eight (48.71%) plants of genotype JS335 SURVIVED.

The article can be accepted after some corrections

Reviewer 4 Report

The manuscript reports on a classic in vitro study for the selection of regenerants from callus and cell suspension culture of soybean, subjected to abiotic stress (in this case, water stress due to the inclusion of PEG in the medium). The work is well structured, correctly described and discussed, accompanied by numerous well-descriptive images, and supported by an adequate bibliography. However, in my opinion, the following major issues must be clarified or better explained before the manuscript can be considered for publication on Plants, i.e .:

  1. Table 2 and Table 3: why a statistical analysis was not applied to better discriminate the obtained results? The Authors reports only the standard deviation, while the application of ANOVA, followed by a mean separation test, would have better discriminated the effect of the different treatments. I can imagine that this choice was due to the fact that means are based on only 5 experimental measurements and the experiment was not repeated a second time, which makes the analysis of the results weak.
  2. Table 2 and Table 3: Authors report that data of Table 2 have been collected after 28-35 days of culture (indicated as 4-5 weeks in the footnote of Table 3). Does it mean that there can be a difference of even 7 days between data collection from the different theses? Why hasn't the survey of the data of the different theses been planned after the same number of days?
  3. The study analyzed the induction of callus from either mature or immature embryonic axes and cotyledons. However, afterwards, the Relative Growth Rates after PEG6000 treatments were calculated mixing the two types of calli (see Table 3). Why? Authors should explain the reason they used different explants for inducing calli, but then they mixed the obtained calli for the PEG6000 trial.

Other minor issues are the following:

  1. Table 1: explain in the caption the composition of the culture media, reported as acronyms (MS2N, MS40D, etc.), without forcing the reader to find their meanings in Materials and Methods.
  2. A concentration of 10 mg/l of 2,4-D was also used in the regeneration trial, in combination with the other PGRs. However, this addition is never commented. Only Table 2 reports CM (Cell Mortality) for the treatments containing 10-mg/l 2,4-D. Authors must give some notes on the effects of such treatment and give details on cell mortality (time, cell/callus morphology, etc.).
  3. Report in Materials and Methods the formula to calculate the Relative Growth Rate. In addition, call it Relative Growth Rate throughout the text (and not sometimes just ‘growth rate’).

Round 2

Reviewer 2 Report

Comments for authors:

Establishment of drought tolerant soybean cultivars is very important.   Authors' screening method of somaclonal variation might be significant.  

However, I cannot see evidences that demonstrate their acquisition of "drought tolerant" plants in this manuscript. I think Title including "drought tolerance" is not appropriate. The candidates (regenerated plantlets) are merely survivors in PEG-containing medium. "Putative tolerant plantlets" in Abstract seems to be overstatement to me as well.

I also need to point out that authors still left mistakes to be amended in the text. For instance, "6000 i e. @ 20% w/v" in Abstract is not proper.

I regret to say but I cannot recommend the version for publication. 

Reviewer 4 Report

------